# On the role of history-dependent adsorbate distribution and metastable states in switchable mesoporous metal-organic frameworks

Francesco Walenszus[1], Volodymyr Bon [1], Jack D. Evans[2], Simon Krause [3], Jürgen Getzschmann[1], Stefan Kaskel [1,4] ✉ & Muslim Dvoyashkin [5] ✉

A unique feature of metal-organic frameworks (MOFs) in contrast to rigid nanoporous materials is their structural switchabilty offering a wide range of functionality for sustainable energy storage, separation and sensing applications. This has initiated a series of experimental and theoretical studies predominantly aiming at understanding the thermodynamic conditions to transform and release gas, but the nature of sorption-induced switching transitions remains poorly understood. Here we report experimental evidence for fluid metastability and history-dependent states during sorption triggering the structural change of the framework and leading to the counterintuitive phenomenon of negative gas adsorption (NGA) in flexible MOFs. Preparation of two isoreticular MOFs differing by structural flexibility and performing direct in situ diffusion studies aided by in situ X-ray diffraction, scanning electron microscopy and computational modelling, allowed assessment of *n*-butane molecular dynamics, phase state, and the framework response to obtain a microscopic picture for each step of the sorption process.

Mesoporous MOFs[1] show ultra-high porosities[2,3] and have demonstrated high efficiency for applications in gas storage[4,5], drug delivery, catalysis and sensors[6]. This ultra-high porosity results in lower bulk moduli than traditional adsorbents almost approaching that of fluids leading to certain flexibility features[7], such as breathing[8], gating[9,10] and even counterintuitive phenomena. One example of a counterintuitive phenomenon is that of negative gas adsorption (NGA)[11], which results from an overloaded metastable state of the porous framework associated with high activation barriers the system cannot easily overcome and that are intrinsic to the solid framework structure, i.e., the constituting nodes and linkers and their barriers to buckle or deform[11]. The

metastability of the overloaded state causes a NGA step in which a characteristic amount of gas, $n_{NGA}$, is expelled at a characteristic pressure, $p_{NGA}$, at a given temperature and gas type along with the contraction from an "open pore" (*op*) to a "contracted pore" (*cp*) state. The huge volume change associated with NGA is associated with high activation barriers for the transformation of the solid phase, which is in essence a heterogeneous solid-solid nucleation[12].

However, in mesopores fluid nucleation is also associated to pronounced activation barriers leading to characteristic hystereses[13,14]. These phenomena are in particular well studied for mesoporous silica like M41S materials which however can be considered as rigid[15–18].

[1]Department of Inorganic Chemistry, Technische Universität Dresden, 01069 Dresden, Germany. [2]Centre for Advanced Nanomaterials and Department of Chemistry, The University of Adelaide, Adelaide, SA 5000, Australia. [3]Nanochemistry department, Max Planck Institute for Solid State Research, 70569 Stuttgart, Germany. [4]Fraunhofer Institute IWS, Winterbergstr. 28, 01277 Dresden, Germany. [5]Institute of Chemical Technology, Universität Leipzig, 04103 Leipzig, Germany. ✉e-mail: stefan.kaskel@chemie.tu-dresden.de; muslim.dvoyashkin@uni-leipzig.de

In the following we analyze the role of fluid diffusion triggering the NGA phenomenon and the structural transition between *op* and *cp* states observed in this series of highly-flexible mesoporous MOF structures[19,20]. Previous studies of NGA using a number of different host-guest combinations, MOFs and adsorbates, addressed in detail the "host"-part, including influence of crystal size[21], structural motifs of metal nodes and linkers (synthesis strategies and post-synthetic modifications)[19,22,23], their micromechanics during sorption processes (in situ powder X-ray diffraction (PXRD)[24], solid-state magic angle spinning NMR[25], density functional theory, in situ EPR[26]) and stress-tests (mercury intrusion)[27], confinement sizes (textural methods, $^{129}$Xe NMR)[28], and extent of flexibility through variation of the framework cell dimensions. The role of adsorbate ("guest"-part) was unraveled using variable pressure- and temperature experiments allowing detailed analysis of pore filling mechanism (in situ neutron powder diffraction (NPD), in situ calorimetry, grand canonical Monte Carlo (GCMC)) and establishing the relation between temperatures at which $n_{NGA}$ is at a maximum to critical temperatures of fluids used as adsorbates[29]. The combined understanding from the above studies has enabled key observations of prerequisites causing the NGA to occur. These include:

the temperature dependence of $n_{NGA}$ indicates this phenomenon scales with the critical temperature ($T_c$) of the fluid[29] suggesting an important role of condensed phase nucleation in providing the out-of-equilibrium stability;

As observed in ref. 30, greater adsorption stresses can be exerted by $CO_2$ sorbates that act to contract the stiffer DUT-46 analogue, a framework hitherto considered as non-deformable in the presence of other adsorbate molecules such as $N_2$, $CH_4$, and *n*-butane, demonstrates the role of fluid-fluid and fluid-host matrix interactions - the stronger these interactions are it becomes more likely that the *op*→*cp* transition will occur and less likely the opposite, i.e., *cp*→*op*;

None of the mesopores (pores with >2 nm in diameter) reach saturation at $p_{NGA}$, their filling by fluid condensation and contraction is amongst the main reasons for NGA transitions. During their filling the host-guest system enters a zone of metastable states of "overloaded" pores, shortly before a contraction is initiated by the cohesive forces of the fluid, resulting overall in the expulsion of fluid from the pores;[31]

in situ NPD and molecular simulation studies have shown that the adsorption-induced stress occurs when the larger pores in the DUT-49 structural motifs begin to fill and metastability of the fluid phase precedes the structural contraction[19].

We hypothesise that these observations, and suggested prerequisites, relate to fluid metastability and variability of its "history"-dependent adsorbate distribution as prerequisites for the occurrence of metastable states along the ad- and desorption isotherms affecting the coupled phase transitions[12]. As a result, the thermodynamic state of nanoconfined fluid for a certain loading becomes dependent on whether it is approached during adsorption or desorption. Until now, their direct experimental evidence remains challenging to obtain and demonstrate. However, pulsed field gradient (PFG) NMR has been revealed to be a decisive technique to identify such diffusion-mediated metastability and variability of fluid configurations in the adsorption hysteresis of mesoporous silica[32]. In this contribution, we back the hypothesis that the fluid nucleation barriers inside mesopores also contribute to the longevity of metastable states and experimentally demonstrate this using in situ PFG NMR[32] combined with in situ PXRD[24] and molecular simulations[33]. This methodological approach was selected for studying sorbate diffusion, its phase state, and the framework response to sorption of *n*-butane simultaneously to gain a possibility for experimental assessment of fluid metastability.

## Results

### Synthesis of archetypical MOFs with different flexibility

The hierarchical porous framework of DUT-149(Cu), isoreticular to DUT-49(Cu) was designed using a ligand with the same length and geometry as it was used for the synthesis of DUT-49(Cu), but containing two methyl groups at the biphenyl moiety (Fig. 1a, b)[34]. Activation of DUT-149(Cu) with supercritical $CO_2$ produces a crystalline powder, showing identical to DUT-49 PXRD and similar crystallite size in SEM (Figs. S4, S5, ESI). However, nitrogen physisorption (77 K) results in a reversible "type Ib" isotherm showing no signs of adsorption-induced transitions. Possessing nearly identical pore morphology and dimensions (Fig. 1d) DUT-49(Cu) and DUT-149(Cu) represent two contrasting materials, where DUT-49 shows an isotropic inelastic unit cell contraction from *op*-state ($a = 46.366(1)$ Å, $V_{UC} = 99742.4$ Å$^3$) to *cp*-state ($a = 36.092(1)$ Å, $V_{UC} = 47351.5$ Å$^3$)[34] but DUT-149 stays in the open form when loaded with *n*-butane at 298 K. Interestingly, a narrow hysteresis is observed while adsorbing *n*-butane on DUT-149(Cu) in the range from 27 kPa to 33 kPa (Fig. 1c, blue isotherm), however no structural changes were observed in PXRD patterns, measured in parallel to physisorption (Fig. S4, ESI). This situation is illustrated in Fig. 1e. Interestingly, similar hysteresis was observed in analogous experiment on DUT-49(Cu), conducted in the pre-NGA pressure range (Fig. S4, ESI). These data suggest history-dependent adsorbate states without changes in the host structures in both cases[12,35].

### Coupling between sorption and molecular dynamics in DUT-149(Cu) and DUT-49(Cu)

Self-diffusion coefficients were experimentally measured under steady state conditions during pressure increase and a subsequent decrease at 298 K using a custom built experimental setup (section 2 of ESI). All measured diffusion attenuation curves exhibit a non-exponential decay due to several possible diffusion regimes occurring simultaneously (see Figs. S1, S2, ESI). These may include diffusion through the gas phase between the crystals, film diffusion on the outer crystal surface, and diffusion upon exchange between the intra- and inter-crystalline spaces. Since an interpretation of attenuation curves and assignment of diffusivities in the entire range of applied gradients would be rather speculative, thus only the high-gradient region resulting in the lowest observable diffusivity is analyzed. Amongst the possible diffusion mechanisms, in-pore diffusion within individual MOF-crystals is expected to possess the lowest value of diffusivity. The resulting diffusivity values of *n*-butane in DUT-149(Cu) as a function of pressure and loading are presented in Fig. 2a, b, respectively.

In the range of loading 9.8-141 mol./u.c. ($\theta_A$) (Fig. 2b, solid symbols), diffusivities increase monotonically and show excellent agreement with data obtained previously for DUT-49(Cu)[37]. A small difference not exceeding 15% is presumably due to the presence of 2,2'-dimethyl-1,1'-biphenyl being part of the window connecting *cub.* and *tet.* pores and hindering diffusion between them, because this is the only structural difference between DUT-149(Cu) and DUT-49(Cu) (see Fig. 1a, b). It is evident from the direct comparison of the calculated pore size distributions of these two MOFs demonstrated in Fig. 1d. According to molecular simulations[37], in this pressure range a gradual filling of *cub.* and *tet.* pores takes place (Figs. S25, S26, ESI). Due to the preferable occupation of the smallest pores, diffusion of *n*-butane is most restricted. An additional factor slowing down the diffusion is accessibility for *n*-butane strong adsorption sites – Cu-metals composing the *cub.* pores as confirmed by in situ NPD measurements[19]. As the loading increases, most of these sites are covered, leading to increased diffusivity for new sorbate molecules.

In situ PXRD patterns, measured at selected points of *n*-butane physisorption isotherm at 298 K, experimentally confirmed this pore filling mechanism. In particular, PXRDs indicate no phase transition to the *cp*-state as it was observed in DUT-49(Cu) (Fig. S6 of ESI). However,

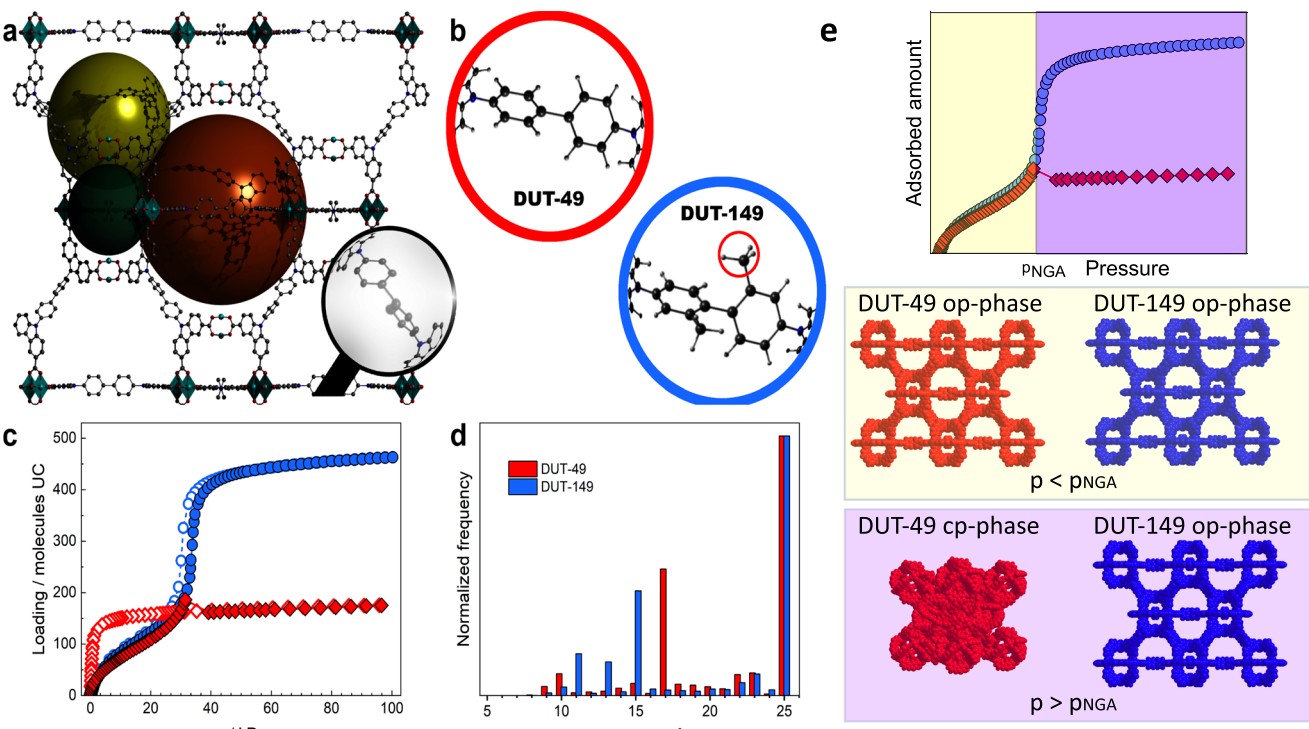

**Fig. 1 | Structure and textural characteristics of DUT-49(Cu) and DUT-149(Cu) MOFs. a** Crystal structure of DUT-49(Cu) and DUT-149(Cu) with highlighted pores (green: cuboctahedral (*cub.*), yellow: tetrahedral (*tet.*), orange: octahedral (*oct.*)). **b** Difference in the spacer part of the linker in DUT-49(Cu) (red) and DUT-149(Cu) (blue). **c** Adsorption/desorption isotherms of *n*-butane at 298 K on DUT-49(Cu) (red) and DUT-149(Cu) (blue). **d** Geometric pore size distribution for DUT-49(Cu) (red) and DUT-149(Cu) (blue) calculated using Zeo + + probe method[36]. The data for DUT-49(Cu) are taken with permission from ref. [37] and shown for comparison. **e** Schematic representation of structural contraction for DUT-49(Cu) undergoing an *op*-state→*cp*-state transition, and structure retaining for DUT-149(Cu) remaining in the *op*-state during *n*-butane physisorption at 298 K.

the Le Bail fit of the PXRD patterns indicates a slight decrease of unit cell parameters, originating form contractive adsorption stress, after DUT-149(Cu) framework exposure to *n*-butane (Fig. 2c). The minimum in the unit cell parameter profile is observed in the pressure range 45–65 kPa in the adsorption branch and 55–40 kPa in the desorption branch of the isotherm. The shift of the minimum in adsorption branch towards higher pressure can be explained by delayed condensation of the n-butane in the mesopore of DUT-149(Cu). These observations are consistent with earlier studies of Gor and co-workers reporting on adsorption-induced deformation in SBA-15 and MCM-41 upon physisorption of *n*-pentane[38] and described the fundamentals of this technique[39]. Rietveld refinement of the selected PXRD patterns confirms the pore filling mechanism in DUT-149(Cu) (Figs. S16–S24, Table S3 of ESI). At the low loadings of 78-112 *n*-butane molecules/UC, the preferable adsorption sites are located in the *cub.* pores and in the pore windows between *cub.* and *tet.* pores. Further increasing of the loading leads to the uniformly continuous filling of *tet.* and *oct.* pores. In situ PXRD patterns, measured on DUT-49(Cu) in parallel to *n*-butane adsorption at 298 K were reported earlier and clearly confirm the contraction of the crystal structure of DUT-49(Cu) at $p_{eq}$ = 35 kPa or *n*-butane loading of ~180 molecules per unit cell[34]. Analysis of in situ PXRDs indicates that at lower loadings DUT-49(Cu) experiences similar adsorption stress and shows decrease of the unit cell parameter from 46.38 Å in vacuum to 46.33 Å at $p_{eq}$ = 32 kPa, which was the last measured point before the structure contracts[34].

Once the maximum of the measured diffusivity is reached at $\theta_B$ (Fig. 2b), continued filling of DUT-149(Cu) with *n*-butane until full saturation at $\theta_C$ = 463 mol./u.c. results in a monotonic decrease of diffusivities, with a gradual slope in comparison to the initial rapid rise. According to GCMC data presented in Fig. 4d, in this range of loadings pore filling has occurred and both *tet.* and *oct.* are filled with *n*-butane.

The inflection point ($\theta_A$), at which diffusivity reaches maximum, precisely coincides with that of adsorption isotherm (~25 kPa) revealing the beginning of capillary condensation process in the *oct.* mesopores of DUT-149(Cu) accompanied by a steep increase of *n*-butane loading in the range 25–40 kPa (Fig. 2b). This observation is associated with the appearance of a condensed phase of *n*-butane inside larger mesopores. This is observed for both DUT-49(Cu) and DUT-149(Cu) indicated by the pore-centered radial distributions[40] plotted in Fig. 3, and is also consistent with a number of diffusion studies dealing with partially filled rigid mesoporous solids (see, e.g.,)[41–43]. The heterogenous nature of the *n*-butane phase in the largest pores (*tet.* and *oct.*) is evident as there is a high density near the pore wall surface and low density in the pore center. In systems featuring mesopores, at the beginning of capillary condensation, the diffusion proceeds partly in the condensed phase (frequently located on the pore surface in the form of a multilayer) and partly in its gaseous counterpart within the pore interior. The progressive pore filling reduces the volume occupied by the gas phase, in which diffusivity is faster compared to the condensed phase, thus leading to lower effective in-pore diffusivities. Also, at higher pressure the density of the gaseous in-pore *n*-butane increases slowing down its diffusion through the gas phase. These two mechanisms contribute to a monotonic decrease of the effective *n*-butane diffusivity until the saturation of pores by a condensed phase of *n*-butane (see data of Fig. 2a in the range of pressures 23–100 kPa and of Fig. 2b in the range of loadings $\theta_A$-$\theta_C$).

Comparison of diffusion data of DUT-149(Cu) (showing no phase transition at 298 K) with DUT-49(Cu) in the range 141–211 mol./u.c. ($\theta_A$-$\theta_B$) (Fig. 2b) remarkably reveals a qualitatively different trend in the loading dependence. The presence of methyl groups of biphenyl moieties in DUT-149(Cu) accessible for *n*-butane diffusing between *tet.-oct.* pores could prohibit a potential increase of diffusivity in this

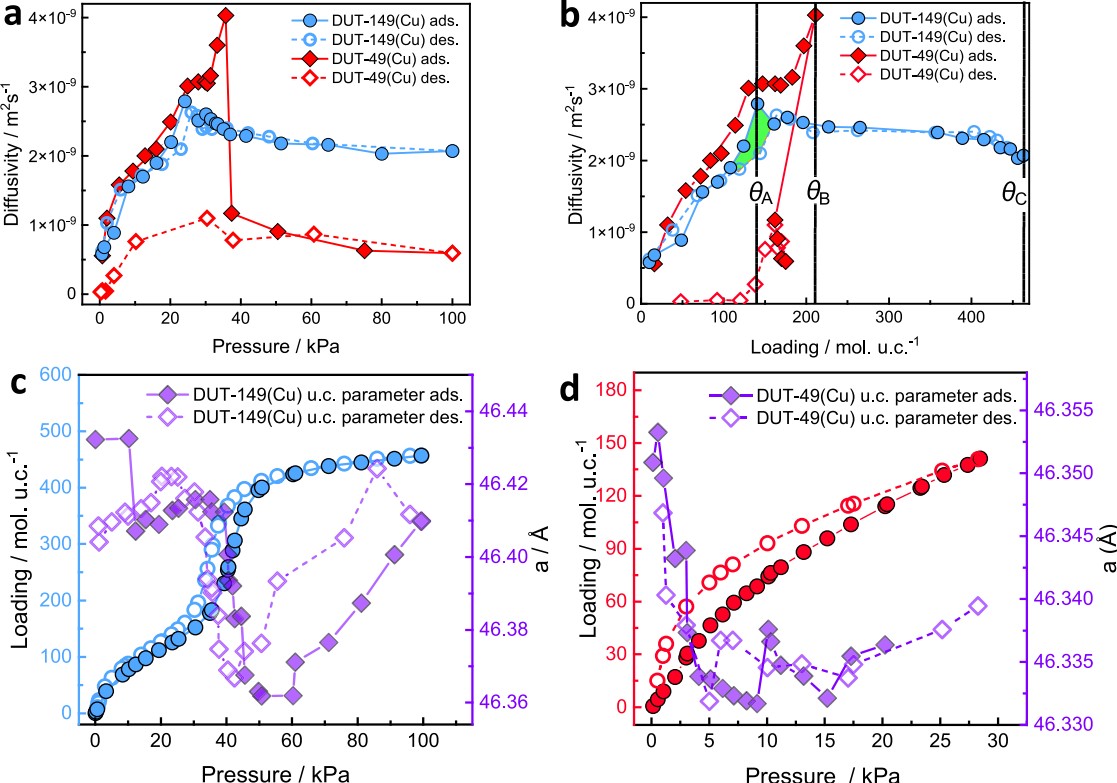

**Fig. 2 | Experimental data for diffusion and adsorption behaviors of *n*-butane of inside pores of rigid DUT-149(Cu) and flexible DUT-49(Cu) MOFs.** Self-diffusion coefficients measured using a high-gradient part of the attenuation curves at 298 K and plotted as a function of pressure (**a**) and loading per unit cell (**b**). The data for DUT-49(Cu) are taken with permission from ref. 37. and shown for comparison. Green shading highlights area of non-equal diffusivities for comparable loadings captured upon sorption and desorption. Adsorption of *n*-butane in DUT-149(Cu) (blue symbols) (**c**) and DUT-49(Cu) (red symbols) (**d**) at 298 K and evolution of unit cell parameters upon adsorption experiment (purple symbols).

range, in contrast to the one observed in DUT-49(Cu) in the same loading range. Since in this range, the loading increases primarily due to the filling of *tet.* and *oct.* pores, the diffusion jumps between these pores contributing to the measured effective diffusivity could potentially prevent its increase due to the progressive occupation of larger pores. Although MD simulations of *n*-butane diffusion in both MOFs do not confirm this hypothesis. The calculations of the pore accessibility indicated that the size of triangular window between *tet.* and *oct.* pores reduces from 13.11 Å in DUT-49(Cu) to 11.92 Å in DUT-149(Cu). Considering the triangular window as a smallest aperture between these pores, the difference in 1.2 Å does not significantly influence the diffusion of *n*-butane molecules between them resulting at the loading of $\theta_B$ in $(8.8 \pm 0.4) \cdot 10^{-9}$ m² s⁻¹ for both structures (Fig. 4b).

Based on results of sorption isotherms obtained by GCMC simulations, the presence of methyl groups on biphenyl linkers facilitates the process of saturation in *tet.* and pore filling in *oct.* pores of DUT-149(Cu) at a lower pressure than in DUT-49(Cu), i.e., at ~23 kPa and ~28 kPa, respectively, suggesting that they might act as additional nucleation centers for condensation (Fig. 4c, d). This range of pressures excellently matches the region $p(\theta_A) \to p(\theta_B)$, where pressure (and loading) dependence of *n*-butane diffusivities in DUT-149(Cu) begins to decrease in contrast to a continued increase observed in DUT-49(Cu). In what follows, the appearance of condensed phase at lower pressure in the former MOF next to linkers reduces interconnectivity between pores containing gas phase throughout the crystals. This process of capillary condensation is clearly visible for *tet.* and *oct.* pores in simulated isotherms in Fig. 4d. This is also supported by in situ PXRD data, namely Rietveld refinement of PXRD patterns, measured at $p_e = 35.5$ kPa or 183 mol. / u.c. The crystal structure indicates five adsorption sites for *n*-butane molecules: two sites with highest

occupancy in *cub.* pore, one site in *tet.*, one in *oct.* and two sites in the window between *tet.* and *oct.* (Figs. S17–S24, Supplementary Movies 1,2, ESI). "Blocking" of the window correspondingly leads to reduction of diffusivity between these pores upon their progressive filling. The impact of stiffness has been also investigated by Grosman demonstrating a systematic small shift of the hysteresis branches to higher $p/p_0$ with reduced porous material stiffness based on porous silicon supported (supposedly stiff) vs. a free-standing membrane (less stiff) and rationalized in terms of thermodynamics[44,45]. In our contribution the apparently "more flexible" DUT-49(Cu) also shows the shift to lower $p/p_0$ in a similar manner wich could be interpreted as an effect of mechanical differences. However, these analyses address the elastic solid deformation (small deformation, reversible), which in our frameworks is apparently comparable and not expected to differ as much based on the observed unit cell changes (For DUT-49(Cu) before the transition). However, the stark mechanical differences of DUT-49(Cu) vs. DUT-149(Cu) concern the inelastic deformation (large deformation, irreversible). In this context the influence of elastic strain on the branch position may be negligible compared to the influence of surface roughness and pore size differences in the two frameworks caused by linker functionalization. This interperation is also in agreement with a follow-up-study by Bossert et al. revising the results of Grosman pointing out the negligible impact of stress and strain on the position of ad- and desorption branches during capillary condensation[46].

The lack of phase transition for DUT-149(Cu) framework compared to its counterpart DUT-49(Cu) additionally contributes to the observed difference of diffusivities, which becomes highly pronounced at loadings approaching $\theta_B$. As expected, the contraction of the DUT-49(Cu) unit cell volume upon NGA event is extensive ($\Delta V_{UC} = 52663.3$ Å³)[34], compared to minor changes in DUT-149(Cu)

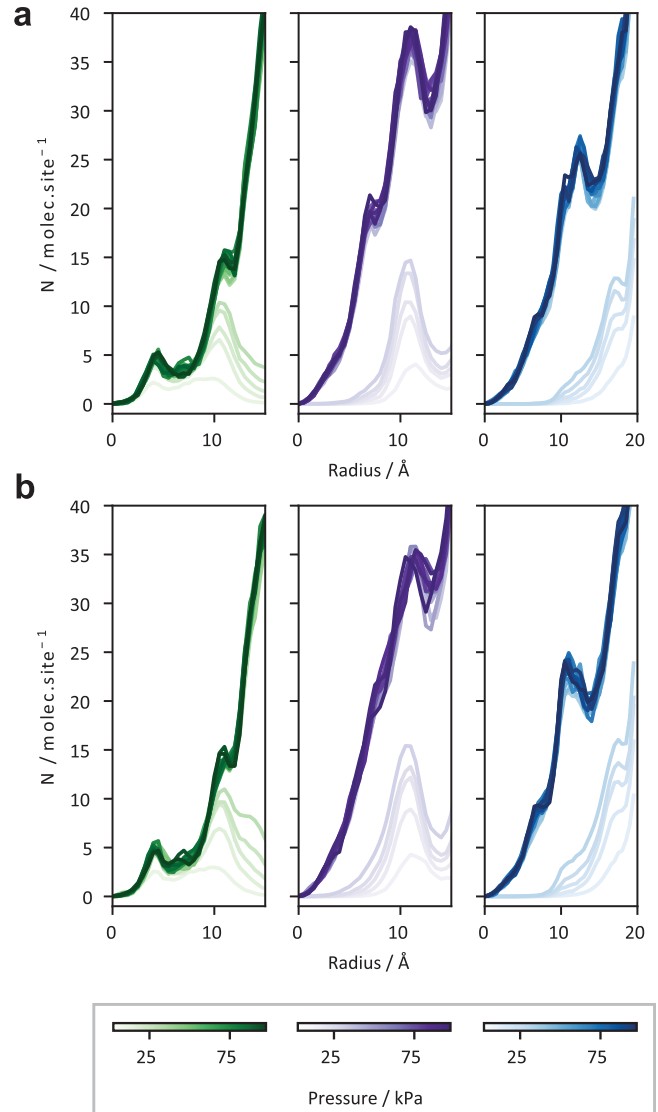

**Fig. 3 | Pore-centered radial distribution functions.** The data are calculated for *n*-butane adsorbed in *cub*. (blue), *tet*. (purple), and *oct*. (green) pores of the DUT-49(Cu) and DUT-149(Cu) 298 K (**a** and **b**, respectively).

$(\Delta V_{UC} = 1452.1 \text{ Å}^3)$ explained by the elastic response to adsorption stress resulting in notably smaller deviations of the unit cell parameter in this range (Fig. 2c). In what follows that despite proximity of pore widths in both MOFs obtained by sorption technique and estimated by PXRD, their exact dimensions at loadings close to the NGA transition might be different, resulting in diffusion through "breathing" confinement in DUT-49(Cu) vs. diffusion within a rigid one in DUT-149(Cu). The latter might also contribute to the observed difference in diffusivities at loadings $\theta_A \rightarrow \theta_B$. To the best of our knowledge, the process of molecular diffusion in temporally variable nanoconfinement comparatively studied together with its rigid counterpart has not been reported previously.

After reaching a complete saturation of pores by *n*-butane at $\theta_C = 463$ mol./u.c., a subsequent decrease of pressure leads to a slight and monotonic increase of diffusivity until 164 mol./u.c. It almost reproducibly repeats the curve measured upon pressure increase.

## Discussion

The most remarkable outcome of the presented results is the diffusion hysteresis (highlighted by green shading in the range of loadings

108–164 mol./u.c. in Fig. 2b), providing experimental evidence for fluid metastability and history-dependent adsorbate distribution during sorption in mesoporous MOFs. This results in the ratio of diffusivities measured upon adsorption and desorption at $\theta_A$ reaching a factor ~1.4 despite equivalent loadings. Utilization of loadings $\theta$, i.e., adsorbed quantity, is necessary when discussing the diffusion hysteresis. This is because similarly looking hysteresis loop may be observed for the $D(p)$ dependence being related to adsorption hysteresis, i.e, with diffusivity expectedly depending on the amounts of adsorbed species (see, e.g.,[47]).

Resulting from the fluid relaxation dynamics in mesopores extensively described by Monson using mean field theory and Monte Carlo approaches[48,49], diffusion hysteresis was experimentally reported for the sorption of cyclohexane in model mesoporous glass and clarified with the help of lattice model simulations in ref. 32. to originate from the ultraslow non-equilibrium dynamics of a fluid redistribution within pore space. Thus, upon equilibration the system undergoes a number of metastable states, in one of which the system is trapped on the experimental time scale and without having a chance to reach equilibrium in any feasible time period[50]. In this consideration, the intrinsic properties of adsorbed fluid play a major role in observed metastability, while the size of confinement created by the porous material and its surface seen by sorbate species remains unchanged throughout the adsorption-desorption cycle. This consideration seems to be applicable also to the DUT-149(Cu) used in this study. Highly-sensitive in situ PXRD data shown in Fig. 2c confirm that distortions of the framework caused by pressure of adsorbed *n*-butane exerted on its surface does not exceed ~1.3% of change of the unit cell parameter in the entire range of pressures used. One of the essential prerequisites of such history-dependent metastability is the presence of liquid-like "droplets" within mesopores since the equilibration dynamics is governed by their spatial redistribution and not by molecular diffusion. The latter is confirmed by manifold equilibration curves measured during a step-like change pressure at loadings $\theta \ll \theta_A$, $\theta_A$ and $\theta \gg \theta_A$ (see data of Fig. S3, ESI).

The broader impact of this diffusion hysteresis for other porous host materials has to be considered with a special care accounting for potential flexibility of the host system. Intrinsic structural flexibility of the material may cause variation of the confinement and significantly influence diffusion. Such a particular case has been prominently demonstrated in our previous study using flexible DUT-49(Cu)[37]. In this case, apparent diffusion hysteresis may not only result from the fluid metastability as described previously, but the effects of variable confinement are superimposed. Another representative example is loading-dependent diffusion of methane in a flexible kerogen matrix, in which absence of mesopores excludes presence of metastable droplets of condensed phase[51], with diffusivities increasing with increasing loading, as opposed to results obtained in the rigid matrix approximation.

To summarize, presented experimental findings clarify and confirm a long-standing hypothesis about the longevity of metastable states upon sorption in mesoporous MOFs and the appearance of a liquid-like condensed phase within mesopores triggering a structural transition in DUT-49(Cu) accompanied by the NGA phenomenon. The analysis of two model materials with almost identical pore structure only differing in inelastic deformability clearly reveals the onset of capillary condensation (and hysteresis) for the rigid system (DUT-149), while the deformable system (DUT-49) at the same point responds with a delayed deformation expelling previously adsorbed molecules leading to NGA. Hence, we attribute fluid nucleation barriers in mesopores to contribute to metastability causing NGA. Although this is a unique demonstration that has become possible due to success in preparing DUT-49(Cu) and DUT-149(Cu), we believe that it is a fundamental observation that can be reproduced for other archetypical MOF-pairs - contracting and not contracting one under certain conditions. However, the role of network topology and pore

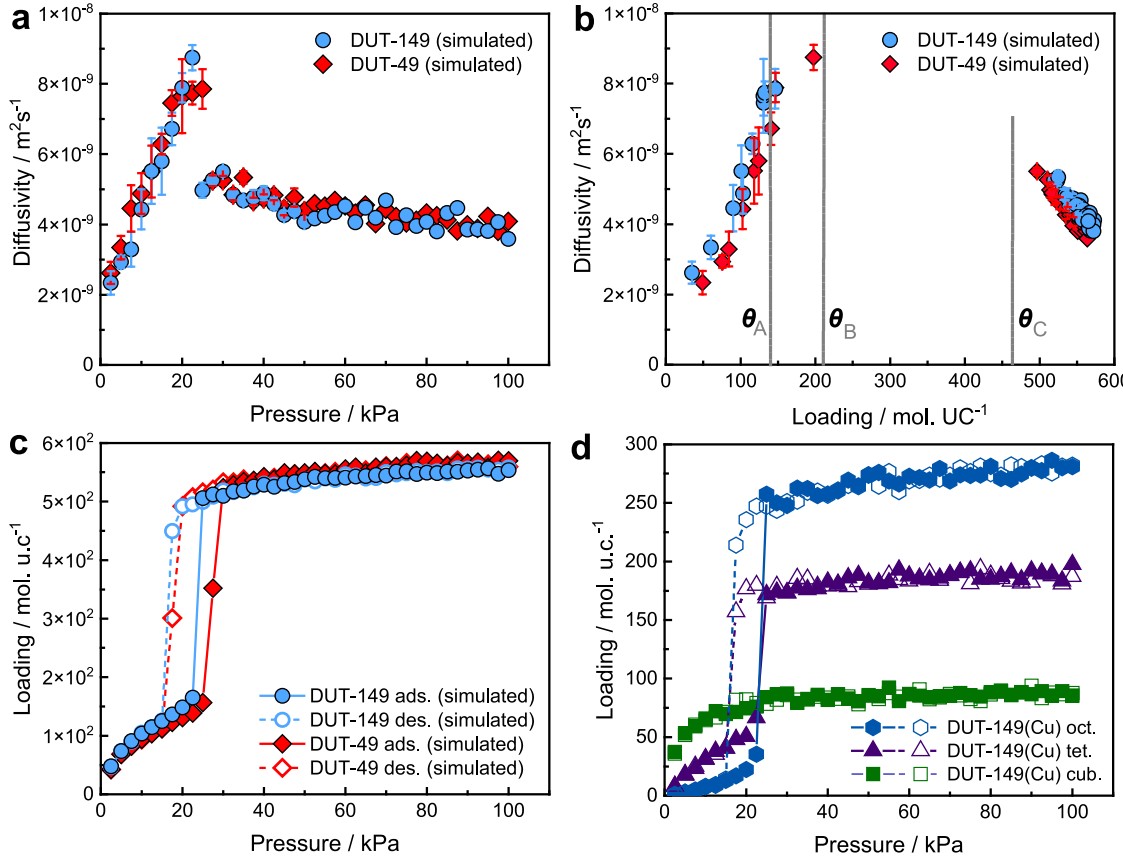

**Fig. 4 | Molecular dynamics- and Monte Carlo simulated diffusion and adsorption processes.** Comparison of self-diffusion coefficients of *n*-butane simulated for an *op*-state of DUT-149(Cu) and DUT-49(Cu) at 298 K and plotted as a function of pressure (**a**) and loading per unit cell (**b**). The dashed vertical line correspond to loading of $\theta_B$ = 211 mol./u.c. **c** Simulated adsorption isotherms in both MOFs and their specific contribution for the each pore type – *cub., tet.*, and *oct.* (**d**).

interconnectivity for NGA may still require further studies. Further loading decrease below 100 mol./u.c. reveals a monotonic decrease of diffusivity until its lowest value is reached at the lowest loading of 9.8 mol./u.c. achieved in the experiment. This is because vast majority of *n*-butane in this range is located in the micropores, for which full reversibility in the adsorption–desorption isotherms is observed (see Fig. 1c in the range of pressures 0–23 kPa) and, thus, the sorption metastable states are not expected.

## Methods
### Synthesis of H₄DBBCDC and DUT-149(Cu)
**Dibutyl 9*H*-carbazole-3,6-dicarboxylate.**

In a 1 l flask 33 g (0.13 mmol) H₂L was suspended in 750 ml 1-butanol and 3 ml of sulfuric acid was added. The mixture was refluxed at 130 °C for 48 h to form a clear yellow solution. The 1-butanol was removed in vacuum, the resulting solid dissolved in chloroform, and extracted with diluted aqueous potassium carbonate solution. The organic phases were combined, dried over MgSO₄ and the solvent removed in vacuum. The obtained solid was recrystallized from ethyl acetate to obtain 45 g (95%) white powder.

$^1$H NMR (600 MHz, CHLOROFORM-*d*) σ (ppm): 1.04 (t, *J* = 7.34 Hz, 6 H) 1.51–1.60 (m, 4 H) 1.81–1.88 (m, 4 H) 4.41 (t, *J* = 6.59 Hz, 4 H) 7.48 (d, *J* = 8.28 Hz, 2 H) 8.19 (dd, *J* = 8.47, 1.69 Hz, 2 H) 8.57 (br. s., 1 H) 8.87 (d, *J* = 1.51 Hz, 2 H).

$^{13}$C NMR (151 MHz, CHLOROFORM-*d*) σ (ppm): 13.83 (s, 1 CH₃) 19.36 (s, 1 CH₂) 30.93 (s, 1 CH₂) 64.82 (s, 1 CH₂) 110.47 (s, 1 CH) 122.75 (s, 1 C) 123.04 (s, 1 CH) 123.14 (s, 1 C) 128.13 (s, 1 CH) 142.66 (s, 1 C) 167.20 (s, 1 C).

MALDI-TOF-MS (m/z): Calculated for C₃₆H₂₀Br₄N₂: 294; found 294 [M-C₄H₉OH]+.

Elemental analysis: Calculated: C: 71.91%; H: 6.86%; N: 3.81%; found: C: 72.01%; H: 6.443%; N: 3.68%.

**Tetrabutyl 9,9'-(2,2'-dimethyl-[1,1'-biphenyl]−4,4'-diyl)bis(9H-carbazole-3,6-dicarboxylate).**

Synthesis conditions: 6.35 g (17.2 mmol) *n*-butylester, 2.5 g (5.76 mmol) 4,4'-diiodo-2,2'-dimethyl-1,1'-biphenyl, 2.5 g (18.1 mmol) potassium carbonate, 430 mg (2.26 mmol) copper(I) iodine, 390 mg (3.39 mmol) *L*-proline, 40 ml DMSO, 105 °C for 24 h and 115 °C for 5 d; Flash column chromatography chloroform: *iso*-hexane: ethyl acetate – 1: 0.06: 0.06 (Rf 0.5); Yield: 3.8 g (72%) white powder.

[1]H NMR (600 MHz, CHLOROFORM-*d*) σ (ppm): 1.05 (t, J = 7.34 Hz, 6 H) 1.51–1.61 (m, 4 H) 1.81–1.90 (m, 4 H) 2.33 (s, 3 H) 4.44 (t, J = 6.96 Hz, 4 H) 7.38–7.48 (m, 1 H) 7.49–7.56 (m, 3 H) 8.14–8.18 (m, 0 H) 8.20–8.25 (m, 2 H) 8.96 (d, J = 1.13 Hz, 2 H).

[13]C NMR (151 MHz, CHLOROFORM-*d*) σ (ppm): 13.64 (s, 1 $CH_3$) 19.18 (s, 1 $CH_2$) 19.97 (s, 1 $CH_3$) 30.74 (s, 1 $CH_2$) 64.66 (s, 1 $CH_2$) 109.68 (s, 1 CH) 122.58 (s, 1 C) 122.81 (s, 1 CH) 122.95 (s, 1 C) 124.18 (s, 1 CH) 127.97 (s, 1 CH) 128.19 (s, 1 CH) 129.21 (s, 1 CH) 136.02 (s, 1 C) 138.14 (s, 1 C) 144.01 (s, 1 C) 166.91 (s, 1 C).

MALDI-TOF-MS (m/z): Calculated for $C_{58}H_{60}N_2O_8$: 839; found 839 [M-$C_2H_5OH$]+.

Elemental analysis: Calculated: C: 76.29%; H: 6.62%; N: 3.07%; found: C: 76.27%; H: 6.432%; N: 2.92%.

**9,9′-(2,2′-Dimethyl-[1,1′-biphenyl]−4,4′-diyl)bis(9*H*-carbazole-3,6-dicarboxylic acid) (H4DBBCDC).**

Synthesis conditions: 3.7 g (4.05 mmol) Tetrabutyl 9,9′-(2,2′-dimethyl-[1,1′-biphenyl]−4,4′-diyl)bis(9H-carbazole-3,6-dicarboxylate), 2.5 g (43.9 mmol) potassium hydroxide, 90 ml THF, 5 ml methanol, 5 ml $H_2O$ + 1 ml $H_2O$ after 1 d, 80 °C for 48 h; Yield: 2.63 g (94%) white powder.

[1]H NMR (500 MHz, DMSO-$d_6$) σ (ppm): 2.28 (s, 3 H) 7.51–7.58 (m, 3 H) 7.58–7.63 (m, 1 H) 7.71 (d, J = 1.89 Hz, 1 H) 8.08–8.17 (m, 2 H) 8.99 (d, J = 1.58 Hz, 2 H) 12.82 (br. s., 2 H).

[13]C NMR (126 MHz, DMSO-$d_6$) σ (ppm): 19.81 (s, 1 $CH_3$) 110.15 (s, 1 CH) 122.58 (s, 1 C) 123.11 (s, 1 C) 123.46 (s, 1 C) 124.30 (s, 1 C) 128.24 (s, 1 CH) 128.39 (s, 1 CH) 131.05 (s, 1 CH) 135.13 (s, 1 C) 138.20 (s, 1 C) 140.42 (s, 1 C) 143.57 (s, 1 C) 167.77 (s, 1 C).

MALDI-TOF-MS (m/z): Calculated for $C_{42}H_{28}N_2O_8$: 687; found 687 [M-H]+.

Elemental analysis: Calculated ($C_{42}H_{28}N_2O_8$ x 2.45 $H_2O$): C: 68.84%; H: 4.53%; N: 3.82%; found: C: 68.44%; H: 4.08%; N: 3.82%.

DRIFT, KBr, 298 K (cm$^{-1}$): 3067 (s, br), 2627 (s), 1904 (s), 1689 (s), 1629 (m), 1601 (s), 1492 (s), 1404 (m), 1293 (m), 1236 (s), 1164 (m), 1135 (m), 1081 (s), 1027 (s), 1008 (s), 906 (m), 824 (m), 771 (s), 737 (s), 725 (m), 699 (s).

**Synthesis and desolvation of DUT-149(Cu).** Due to the reduction of $Cu^{2+}$ by NMP for longer reaction times, the synthesis of large-scale MOF powder of DUT-149(Cu) was conducted in DMF. The reaction was carried out in a round bottom flask, and the reaction mixture was stirred and heated at 80 °C using an oil bath. All reactions were conducted in the following manner: 500 mg (0.73 mmol) H4DBBCDC linker was dissolved in 130 ml of DMF; 8.3 mL (145 mmol) of acetic acid were added and the solution was stirred at 80 °C. To this solution 438 mg (1.81 mmol) of Cu(NO$_3$)$_2$·3H$_2$O were added and dissolved, the flask was closed and the reaction mixture stirred for 96 h at 80 °C. Afterwards the MOF powders were separated from the mother liquid via centrifugation and washed with fresh DMF for at least three times over 3 days. Afterwards the solvent was replaced by anhydr. acetone in multiple washing cycles (at least 6 times over 3 days). The ethanol/acetone suspended MOF powder was placed in glass filter frits in a Jumbo Critical Point Dryer 13200 J AB (SPI Supplies) which was subsequently filled with liquid $CO_2$ (99.995% purity) at 288 K and 5 MPa. To ensure a complete substitution of the solvent by $CO_2$, the liquid in the autoclave was exchanged with fresh $CO_2$ at least 18 times over a period of 5 days using a valve at the bottom of the autoclave. The temperature

and pressure were then risen beyond the supercritical point of $CO_2$ to 308 K and 10 MPa and kept until the temperature and pressure was constant. The supercritical $CO_2$ was steadily released over 3 h and the dry powder was transferred and stored in an argon filled glove box. To ensure complete removal of the solvent (especially from the open metal sites of the Cu-paddle-wheels) additional activation at 423 K in a Schlenk-tube under dynamic vacuum of 10$^{-4}$ kPa for at least 24 h was performed.

Elemental analysis: Calculated ($Cu_2C_{42}H_{20}N_2O_8$): C: 62.14%; H: 2.98%; N: 3.45%; found: C: 61.22%; H: 3.154%; N: 3.5%.

**In situ pulsed field gradient NMR**
In situ NMR diffusion studies at controlled *n*-butane loadings were performed at 298 K by [1]H PFG NMR using a home-built NMR spectrometer with a magnetic field of 2.35 T operating at 100.13 MHz resonance frequency and applied gradients up to 22 T/m. The home-designed PEEK cylindrical in situ NMR cell (height 120 mm, diameter 7.6 mm) was filled with 43.3 mg of desolvated DUT-49(Cu) and connected to the low-pressure port of BELSORP-max volumetric adsorption instrument by 6 m long stainless-steel tube (d = 1/8 inch). DUT-149(Cu) was transferred into the cell under the inert gas conditions. Before the experiment, DUT-149(Cu) was degassed for 1 h in ultrahigh vacuum (p < 0.1 Pa). Further the controlled *n*-butane pressures were adjusted using the needle valves of the gas dosing system. Each loading was equilibrated at least 15 min before the PFG NMR measurement. The constant temperature of 298 K during the experiment was assured using PT100 sensor integrated into the probe.

Stimulated echo pulse sequence was used in all of the diffusion experiments. The application of 13-interval pulse sequence was also checked. There was no difference in obtained diffusion decays by 13-interval pulse sequence from the one obtained by stimulated echo pulse sequence within the experimental uncertainty of obtained signal-to-noise ratios. Thus, the latter pulse sequence was used. The parameters for diffusion experiments were: $\triangle$ = 10 ms, $\delta$ = 0.8 ms, and $\tau$ = 1.2 ms, where $\triangle$ is the time between gradient pulses, $\delta$ is the gradient pulse duration, $g$ is the gradient field strength and $\tau$ is the time between the first and second ($\frac{\pi}{2}$) pulses. The diffusion attenuations have been fitted with the single-exponential function as shown by dashed lines in the Figs. S1, S2, ESI from which self-diffusion coefficients have been extracted. The values of experimental uncertainties lie within the size of the symbols. It is valid for all loadings, since at lower loadings, more signal accumulation was applied to obtain signal-to-noise levels consistent with other measurements at higher loadings. To obtain uncertainties, the least-squares fitting to a monoexponential function in the range of high gradients corresponding to the slowest diffusivity was applied with the cut-off value resulting in errors not exceeding the standard deviation.

**In situ PXRD in parallel to adsorption**
In situ PXRD patterns on DUT-49(Cu) and DUT-149(Cu) in parallel to *n*-butane physisorption at 298 K were measured using home-built dedicated instrumentation, based on Empyrean powder X-ray diffractometer (ω−2θ goniometer, alpha1 system) using a customized setup based on ARS DE-102 closed cycle helium cryostat (T = 30–300 ± 0.1 K) and adsorption cell, based on 1.33″ CF-flange and Beryllium dome. The cell was connected to the low-pressure port of the BELSORP-max (Microtrac MRB) volumetric adsorption instrument. The TTL-trigger was used for establishing the communication between BELSORP-max and Data Collector software and ensure the measurement of adsorption isotherm and PXRD patterns in fully automated mode. The diffraction experiments were performed using ω−2θ scans in transmission geometry in the range of 2θ = 2–50°. Parallel beam optics (hybrid 2xGe(220) monochromator, 4 mm mask, primary divergence and secondary antiscatter slits with ¼° opening) was used for the data collection. Pixcel-3D detector in 1D scanning mode (255

active channels) was used for recording of the scattered intensities. A physisorption of *n*-butane at 298 K was measured on 8.9 mg of DUT-149(Cu) sample, mounted in the X-ray beam, and PXRD patterns were recorded after equilibration (0.1% of pressure change within 300 s) at selected points of the isotherm. Adsorption and desorption isotherm, measured in situ and corresponding PXRD patterns are given in the Figs. S4a, b, ESI.

PXRD patterns were analyzed using Le Bail profile fit procedure, integrated in Fullprof software. The zero-line shift was not refined in order to get exact changes in unit cell parameters. Extracted unit cell parameters, combined in the same plot with adsorption and desorption isotherms reflect the adsorption stress, the framework subjected to upon adsorption and desorption of the *n*-butane molecules in the mesopores (Fig. S5a–c, ESI). In adsorption branch of the isotherm, the strongest contraction of the unit cell axes from 46.42 Å to 46.36 Å is observed in the pressure range of 40–50 kPa. In the pressure range of 60–100 kPa, linear increase of unit cell parameter from 46.36 to 46.41 Å indicate the release of the stress because of the complete filling of the pores with *n*-butane. In desorption branch, unit cell constant decreases in the pressure range 85–42 kPa, reaching the minimum of 46.365 Å. Further desorption leads to reducing the stress, reflected in the increase of the unit cell constant from 46.365 Å to 46.425 Å at 20 kPa. Interestingly, comparison of the unit cell parameter profiles shows the shift of the minimum unit cell parameter, reflected maximum adsorption stress, from 50 to 60 kPa in adsorption branch, to 40–50 kPa in desorption branch. This can be referred to the condensation of the n-butane in the mesopore of the MOF, and explain the resulting hysteresis in the isotherm.

SEM images were acquired from the both samples before and after in situ PXRD experiment in order to evaluate the crystal surface before and after gas physisorption experiments (Figs. S4c, d, S5d, e, ESI). The images indicate no strains and boundaries on the crystal surfaces, which would indicate the macroscopic changes upon adsorption and desorption process.

### Pore filling in DUT-149(Cu) by Rietveld refinement

In order to localize the adsorbed *n*-butane molecules in the pores, selected PXRD patterns were subjected to Rietveld analysis. The crystal structure of DUT-149(Cu) (space group $Fm\bar{3}m$) was used as initial model for refinement. PXRD patterns, measured on degassed solid fits well to the model. One oxygen atom per paddle-wheel, located within the smallest cuboctahedral pore of the structure was remained after the desolvation procedure. The *n*-butane loadings selected for refinement are listed in the table S1. The *n*-butane molecules were added in the pores of DUT-149(Cu) and treated as rigid bodies in the refinement, allowing the translation and rotation of the molecule in the pores. The occupancy of *n*-butane molecules was refined without constraints. The Rietveld plots for all selected loadings are given in Figs. S7–S15, ESI, and the corresponding convergence factors and number and occupancies of *n*-butane molecules are listed in the Table S2. CCDC-2247296-2247304 contain the supplementary crystallographic data for DUT-149(Cu) empty framework and crystal structures with defined n-butane loadings. These data can be obtained free of charge from the Cambridge Crystallographic Data Centre via www.ccdc.cam.ac.uk/data_request/cif.

### Monte Carlo and molecular dynamics simulations

The force field for the copper paddlewheels was taken from previous work of Vanduyfhuys et al.[52]. The QuickFF[53] program was then used to derive a consistent force field for the organic linker of the DUT-49(Cu) and DUT-149(Cu) framework. The representative ligand was geometrically optimized and the Hessian in equilibrium was calculated with the B3LYP[54–56] functional and the cc-pVDZ basis set as described in the ORCA program[57]. The covalent force field consists of harmonic bond,

bend and out-of-plane distance contributions, single cosine dihedrals and cross terms describing the coupling between neighboring bonds as well as the angles and their constituting bond. The electrostatics were described using the gaussian-spread atomic charges computed using the MBIS scheme[58]. The electrostatic contributions were supplemented with van der Waals interactions taken from the MM3 force field[59]. The TraPPE united-atom model was used, for the n-butane adsorbates, in which the Lennard-Jones parameters were rescaled to be used with the MM3 potential, by requiring the potential depth and distance of the minimum to be the same. This is done in order for consistency with the MM3 potential used for the framework as used for previous studies[60].

Adsorption was simulated using grand canonical Monte Carlo (GCMC) simulations as employed by the YAFF package (version 1.6.0.post21). To produce representative isotherms simulations were performed for 41 values of gas pressure at 298 K. The chemical potential at each temperature and gas pressure was calculated with the Peng–Robinson equation of state[61]. Monte Carlo steps of insertion, deletion, translation and rotation were used in equal probability and $1 \times 10^7$ steps were simulated and the final $5 \times 10^6$ steps used to compute the average properties. Interactions were treated by the Lenard-Jones potential described above. Desorption simulations were simulated by begin the GCMC routine from a prior simulation snapshot at saturation (1 bar).

Molecular dynamic simulations in the NPT ensemble were simulated using the lammps simulation package[62]. Simulations used a time step of 0.5 fs and the Nosé-Hoover thermostat and barostat with a thermostat relaxation time of 100 fs and barostat relaxation time of 1000 fs, set to 298 K and 0 atm. Initial adsorbate positions were taken from snapshots of the GCMC simulations. Diffusion was computed from the slope of mean-squared displacement using the final 1 ns of a 2 ns trajectory using the MDanalysis package[63,64]. For pressures <0.5 bar four unique trajectories were produced, for each loading, to determine the average and variance of diffusion.

Representative input files and further data can be found on the Github repository of Jack D. Evans, https://github.com/jackevansadl/supp-data.

## Data availability
All data are available in the main text and the Supplementary information. In addition, the simulation data are accessible through the GitHub repository under https://github.com/jackevansadl/supp-data/tree/master/XX-Walenszus. CCDC-2247296-2247304 contain the supplementary crystallographic data for DUT-149(Cu) empty framework and crystal structures with defined *n*-butane loadings. These data can be obtained free of charge from the Cambridge Crystallographic Data Centre via www.ccdc.cam.ac.uk/data_request/cif.

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

## Acknowledgements

This project has received funding from the European Research Council (ERC) under the European Union's Horizon 2020 research and innovation programme (grant agreement No. 742743). V.B. and S.Ka. thank the BMBF (Projects No. 05K22OD1 and No. 05K22OD2) for financial support. J.D.E. is the recipient of an Australian Research Council Discovery Early Career Award (project number DE220100163) funded by the Australian Government. Phoenix HPC service at the University of Adelaide are thanked for providing high-performance computing resources. This project was undertaken with the assistance of resources and services from the National Computational Infrastructure (NCI), which is supported by the Australian Government. S. Kr. Acknowledges financial support by the Carl-Zeiss-Stiftung. M.D. acknowledges the support from the German Research Foundation (DFG), grant DV 58/2-2. The PFG NMR diffusion experiments have been performed in the Felix Bloch Institute for Solid State Physics, Division of Applied Magnetic Resonanse, Leipzig University. Technical assistance of Stefan Schlayer is gratefully acknowledged. Nadine Bönisch is acknowledged for acquiring SEM images.

## Author contributions

S.Ka. and M.D. conceptualized and designed the study. S.Kr. conducted the synthesis the performed the characterization of DUT-149(Cu) by PXRD and nitrogen physisorption (77 K) and n-butane (273 K). F.W. conducted the synthesis and performed the characterization of DUT-49(Cu). F.W., V.B and M.D. designed and conducted in situ PFG NMR experiments on DUT-149(Cu). M.D. analized and interpreted in situ PFG NMR data. V.B. conducted in situ PXRD experiments on DUT-49(Cu) and DUT-149(Cu) and conduted Le Bail fit. J.G. performed Rietveld analysis of in situ PXRD data. J.D.E. performed geometrical porosity calculations, MD and GCMC simulations on DUT-149(Cu). M.D. wrote the initial draft of the paper. All authors contributed to the refining and improving of the final draft.

## Competing interests

The authors declare no competing interests.
