## [Peer Review File · Nature Communications]

On the role of history-dependent adsorbate distribution and metastable states in switchable mesoporous metal-organic frameworksREVIEWER COMMENTS

Reviewer #1 (Remarks to the Author):

This paper focuses on studies of changes of sorbate diffusion and structure in a flexible MOF upon increasing sorbate loading pressure. The results obtained for the flexible MOF are compared with those obtained for the isorecticular MOF lacking such framework flexibility. Owing to the successful synthesis of such two MOFs this study is unique and offers significant new observations and insights, such as an observation of the diffusion hysteresis and its discussion. However, the following questions and comments need to be addressed before the publication is considered.

Comments and questions:

P6: Since the largest pore size of around 2.4 nm is only around 20% larger than the largest size of micropores (2 nm) the discussion of sorbate "phase" in 2.4 nm pores might not be very meaningful because majority of sorbate molecules can experience interactions with pore walls at any particular time. This is the reason why "sorbate phase" such as "gas phase" and "condensed phase" are not usually discussed for sorbates located inside micropores. Molecular level simulations might clarify this issue.

Lines 178-180: Results of MD simulations were discussed with no reference to details of such simulations.

Line 244: It is indicated that details of MD simulations are presented in the Supporting Information file. This reviewer, however, could not find such details in the submitted file.

Line 207: I suggest adding the value from literature for "the contraction of the DUT-49(Cu) unit cell". It is not presented in Fig. 2.

Line 222: What are the experimental uncertainties for both concentration and diffusivity values in the area highlighted by green shading in the range of loadings 108-164 mol./u.c. in Fig. 2b? How were these uncertainties estimated?

P.9: I suggest providing a more detailed discussion of the diffusion hysteresis observed for DUT-49(Cu).

Section 2 in Supporting Information: In order to confirm that the reported diffusivities correspond to intra-crystalline diffusion the authors should compare the root mean square displacements under the PFG NMR measurement conditions with the MOF crystal sizes.

Figs. S1, S2: I am concerned that very different PFG NMR attenuation ranges were used to obtain diffusivities for different loading pressures. Is there some evidence or consideration indicating that all these different ranges correspond to the same type of diffusion, i.e. intra-crystalline diffusion, which is not perturbed by crystal boundaries?

Vague/unclear phrases:

Lines 35,36, "at a given temperature and gas..." Do you mean gas type or gas concentration?

Line 156 "coexisting with its gas phase either within the same mesopore or with neighboring pores."

Reviewer #2 (Remarks to the Author):

The work titled "On the role of history-dependent adsorbate distribution and metastable states in switchable mesoporous metal-organic frameworks" by Dvoyashkin reported the adsorption mechanism of butane within two related MOFs, DUT-49 and DUT-149. The author applied various experimental techniques, particularly using in situ PFG NMR combined with in situ PXRD and molecular simulations, to reveal the diffusivity of butane within the pores. Although new understanding of "diffusion hysteresis" were provided, this study is more like an add-on to the previously published work "Molecular Diffusion in a Flexible Mesoporous Metal–Organic Framework over the Course of Structural Contraction" (J. Phys. Chem. Lett. 2020, 11, 9696–9701) by the same group where similar techniques were applied to study similar properties of the same adsorbate (n-butane) in one of the MOFs for this current study (DUT-49), hence, I feel the scientific novelty of this work is not reaching the requirement for publication on Nat. Comm.

Reviewer #3 (Remarks to the Author):

The manuscript presents a combination of experimental and molecular simulation studies of adsorption and diffusion of hexane in two MOF structures, and relate it to each other and to the structural change of the MOFs. One of the main results is the hysteresis of the diffusivity as a function of amount adsorbed, which according to the authors suggest the fluid metastability during sorption in MOFs.

The experimental results are novel and interesting for the adsorption and porous materials community. However, I feel that the manuscript could have provided a more thorough analysis of the data to draw stronger conclusions, rather than stating that "deeper understanding of fundamental differences in contributing diffusion mechanisms leading to observed deviations requires further investigation."

I also believe that some of the aspects of this work are not specific to MOF materials, and could be discussed in the context of the broader literature relating adsorption to metastability of fluid and to deformation.

For example:

1. Grosman, A.; Ortega, C. Influence of elastic strains on the adsorption process in porous materials: an experimental approach. *Langmuir* 2009, 25, 8083–8093.
2. Grosman, A.; Ortega, C. Cavitation in Metastable Fluids Confined to Linear Mesopores. *Langmuir* 2011, 27, 2364–2374.
3. Bossert, M., Grosman, A., Trimaille, I., Noûs, C. and Rolley, E., 2020. Stress or strain does not impact sorption in stiff mesoporous materials. *Langmuir*, 36(37), pp.11054-11060.

The manuscript mentions MCM41 materials as "rigid", while Fig 2 shows the same order of magnitude of strain during adsorption of alkanes as was previously shown for alkanes in MCM41/SBA15 e.g. in:

4. Gor, G.Y., Paris, O., Prass, J., Russo, P.A., Ribeiro Carrott, M.M.L. and Neimark, A.V., 2013. Adsorption of n-pentane on mesoporous silica and adsorbent deformation. *Langmuir*, 29(27), pp.8601-8608.

and not only the magnitude but also the trend similar to other non-MOF materials:

5. Gor, G.Y., Huber, P. and Bernstein, N., 2017. Adsorption-induced deformation of nanoporous materials—a review. *Applied Physics Reviews*, 4(1), p.011303.

The manuscript also lack the details on simulations. It says that Supporting Info has in particular "Details of the ... MD simulations", but it does not. I recommend that not only the details of modeling methods are included in the PDF, but also all the main input files are included, so that the study could be easily reproduced if needed.

Additionally, I have a few minor comments:

1. The abstract has a typo "soprtion"
2. The role of Fig 1e (top) is unclear, furthermore this figure is not mentioned in the text.
3. Size of the panels in Fig 2 are not the same
4. Different units are used for pressure in Fig 2 and Fig 4
5. Figures S7-S15 look like screenshots of very poor quality, I would suggest using the data and preparing the plots which meet the standards of the journal

RESPONSE TO REVIEWERS' COMMENTS

Reviewer #1 (Remarks to the Author):

This paper focuses on studies of changes of sorbate diffusion and structure in a flexible MOF upon increasing sorbate loading pressure. The results obtained for the flexible MOF are compared with those obtained for the isorecticular MOF lacking such framework flexibility. Owing to the successful synthesis of such two MOFs this study is unique and offers significant new observations and insights, such as an observation of the diffusion hysteresis and its discussion. However, the following questions and comments need to be addressed before the publication is considered.

We are very grateful for such a positive evaluation of our work and for the suggestions helping further improve the manuscript. The revisions made are described below in detail and or colour highlighted.

Comments and questions:

P6: Since the largest pore size of around 2.4 nm is only around 20% larger than the largest size of micropores (2 nm) the discussion of sorbate "phase" in 2.4 nm pores might not be very meaningful because majority of sorbate molecules can experience interactions with pore walls at any particular time. This is the reason why "sorbate phase" such as "gas phase" and "condensed phase" are not usually discussed for sorbates located inside micropores. Molecular level simulations might clarify this issue.

Indeed, the similar size and connected nature of the tetrahedral (*tet.*) and octahedral (*oct.*) pores lead them to behave in a similar fashion, as observed in the pore-centered radial distribution functions in Figure 3. However, this does suggest a heterogeneous phase is present in these larger pores (*tet.* and *oct.*). The sharp peak in distribution close to the pore surface at pressures below the condensation point is evidence of the "condensed phase" and density of ~ 0 molecules per site at the center of the pore is indicative of "gas phase". This phase separation is not observed in the smallest pore (*cub.*). We have edited our discussion to improve the introduction of Figure 3 and included supplementary figures demonstrating the adsorption location of *n*-butane in each pore during the adsorption isotherm.

Line 132, added text: "(Supplementary Figures 25-26)"

Lines 162-164, added text: "The heterogeneous nature of the *n*-butane phase in the largest pores (*tet.* and *oct.*) is evident as there is a high density near the pore wall surface and low density in the pore center. In systems featuring mesopores..."

Lines 178-180: Results of MD simulations were discussed with no reference to details of such simulations.

Line 244: It is indicated that details of MD simulations are presented in the Supporting Information file. This reviewer, however, could not find such details in the submitted file.

We apologize for failing to add the details of the GCMC and MD simulations. The relevant details have been added to the supporting information as a separate section 4 and representative input files can be found at the GitHub repository: <https://github.com/jackevansadl/supp-data/tree/master/XX-Walenzsus>. The code-free link is provided in the "DATA AVAILABILITY" section (lines 283-286). Additionally, we have provided the supplementary crystallographic data for DUT-149 empty framework and crystal structures with defined *n*-butane loadings in this section (lines 286-288).

Line 207: I suggest adding the value from literature for “the contraction of the DUT-49(Cu) unit cell”. It is not presented in Fig. 2.

The literature value on the unit cell contraction of DUT-49(Cu) upon NGA are now added and compared to the minor changes in DUT-149(Cu) caused by adsorption stress. The values are discussed in the text (lines 88-89, lines 225-226).

Lines 88-89, added text: “shows an isotropic inelastic unit cell contraction from *op*-state ($a = 46.366(1) \text{ \AA}$, $V_{UC} = 99742.4 \text{ \AA}^3$) to *cp*-state ($a = 36.092(1) \text{ \AA}$, $V_{UC} = 47351.5 \text{ \AA}^3$)³⁴”

Lines 225-226, added text: “DUT-49(Cu) unit cell volume upon NGA event is extensive ($\Delta V_{UC} = 52663.3 \text{ \AA}^3$)³⁴ compared to minor changes in DUT-149(Cu) ($\Delta V_{UC} = 1452.1 \text{ \AA}^3$)”

Line 222: What are the experimental uncertainties for both concentration and diffusivity values in the area highlighted by green shading in the range of loadings 108-164 mol./u.c. in Fig. 2b? How were these uncertainties estimated?

In the highlighted area and at higher loadings, the values of experimental uncertainties lie within the size of the symbols. It is valid for all loadings, since at lower loadings, more signal accumulation was applied to obtain signal-to-noise levels consistent with other measurements (at higher loadings). To obtain uncertainties, the least-squares fitting to a monoexponential function in the range of high gradients corresponding to the slowest diffusivity was applied with the cut-off value resulting in errors not exceeding the standard deviation. This clarification is now added to Section 2 of ESI describing details of NMR experimentation. The uncertainties for concentration values were deduced from the sorption isotherms shown in Fig. 1c.

P.9: I suggest providing a more detailed discussion of the diffusion hysteresis observed for DUT-49(Cu).

On pages 10 and 11, the related discussion is extended by adding relevant details in the context of diffusion hysteresis, also addressing its broader impact. New text is added in the lines between 242-267.

Section 2 in Supporting Information: In order to confirm that the reported diffusivities correspond to intra-crystalline diffusion the authors should compare the root mean square displacements under the PFG NMR measurement conditions with the MOF crystal sizes.

The supplementary Table S1 summarizing the resulting self-diffusion coefficients obtained from the fitting of attenuation curves of Figs. S1, S2 and calculated root mean squared displacements (RMSD) is added to Section 2 of ESI. All RMSD values are smaller than the average crystal size of $\sim 20 \mu\text{m}$ resulting in the major contribution of the intra-crystalline diffusion. We performed an additional SEM-characterization to evaluate average crystal size and added relevant images to Fig. S4 (d-f) on page 9 of ESI

Figs. S1, S2: I am concerned that very different PFG NMR attenuation ranges were used to obtain diffusivities for different loading pressures. Is there some evidence or consideration indicating that all these different ranges correspond to the same type of diffusion, i.e. intra-crystalline diffusion, which is not perturbed by crystal boundaries?

It is a very fair concern. Unfortunately, due to applied “diffusometry-type” of the PFG NMR technique in this work, we are lacking the spectroscopic information. It would have been great to perform the in situ DOSY studies (i.e., with spectroscopic information included), where the change in the chemical shift of measured signals could give a hint whether molecules diffuse within the MOF-crystals only during the entire diffusion time ($\sim 10 \text{ ms}$) or do escape the intra-crystalline space and

crossover to the long-range diffusion. To the best of our knowledge, such experiments are technically very difficult, as they would require combination of high field and high gradients, such as, e.g., in the National High Magnetic Field Laboratory Facility in Gainesville FL, and which we do not have. In this context, we have no direct experimental evidence of this. However, given that estimated average crystal size is larger than the root mean square displacements and taking into account nearly defect-free crystal structure confirmed by in situ XRD studies, we do not expect any significant contribution of diffusion perturbed by crystal boundaries. Regarding this concern, authors remain open for a potential future cooperation to gain a possibility of accessing the spectroscopic information in diffusion experiments, which will be highly appreciated.

Vague/unclear phrases:

Lines 35,36, “at a given temperature and gas...” Do you mean gas type or gas concentration?

It meant to be a “gas type”, since the contraction conditions may vary depending on the type of gas used. The typo in the line 36 is corrected.

Line 156 “coexisting with its gas phase either within the same mesopore or with neighboring pores.” Corrected. This part indeed looks misleading and therefore was removed.

Reviewer #2 (Remarks to the Author):

The work titled “On the role of history-dependent adsorbate distribution and metastable states in switchable mesoporous metal-organic frameworks” by Dvoyashkin reported the adsorption mechanism of butane within two related MOFs, DUT-49 and DUT-149. The author applied various experimental techniques, particularly using in situ PFG NMR combined with in situ PXRD and molecular simulations, to reveal the diffusivity of butane within the pores. Although new understanding of “diffusion hysteresis” were provided, this study is more like an add-on to the previously published work “Molecular Diffusion in a Flexible Mesoporous Metal–Organic Framework over the Course of Structural Contraction”(J. Phys. Chem. Lett. 2020, 11, 9696–9701) by the same group where similar techniques were applied to study similar properties of the same adsorbate (n-butane) in one of the MOFs for this current study (DUT-49), hence, I feel the scientific novelty of this work is not reaching the requirement for publication on Nat. Comm.

We are thankful to reviewer for the pointing out the novelty issue. Indeed, we published the paper reporting on the diffusivity study in DUT-49(Cu) upon *n*-butane physisorption. The main idea of the paper was to show how the change of confinement (contraction of mesopore from 2.4 nm to 1.0 nm) will change the self-diffusion of *n*-butane in the framework. We could also monitor the diffusivity in the metastable range just before the contraction. However, the most important fundamental questions related to fluid metastability and history-dependent adsorbate states could not be answered because of the phase transition.

Here, we succeeded in the design of the isorecticular framework DUT-149(Cu) with very similar composition and pore system, which, however, behave as a rigid system in the same experimental conditions. Therefore, we could monitor the self-diffusivity of *n*-butane upon adsorption and desorption in entire loading range.

Concluding, we are still convinced that current study is unique and would like to argue as follows:

- 1) To the best of our knowledge, our paper on DUT-49(Cu) (J. Phys. Chem. Lett. 2020, 11, 9696–9701) is the only work reporting experimental diffusivity on mesoporous framework undergoing a structural transition by PFG NMR.

- 2) Authors are not aware about the diffusion studies on two porous solids showing identical pore system, pore volume, crystal size, similar chemical composition, but only one of them shows adsorption induced contraction. Diffusion values, obtained at different loadings and pressures upon adsorption and desorption of *n*-butane, and particularly hysteresis obtained between adsorption and desorption provides clear insights into the fundamental origin of structural contraction and NGA from the prospective of the probe molecule.

Reviewer #3 (Remarks to the Author):

The manuscript presents a combination of experimental and molecular simulation studies of adsorption and diffusion of hexane in two MOF structures, and relate it to each other and to the structural change of the MOFs. One of the main results is the hysteresis of the diffusivity as a function of amount adsorbed, which according to the authors suggest the fluid metastability during sorption in MOFs.

The experimental results are novel and interesting for the adsorption and porous materials community. However, I feel that the manuscript could have provided a more thorough analysis of the data to draw stronger conclusions, rather than stating that "deeper understanding of fundamental differences in contributing diffusion mechanisms leading to observed deviations requires further investigation."

We also highly appreciate a positive evaluation of our contribution by this Reviewer and his/her suggestions which we readily address as follows.

I also believe that some of the aspects of this work are not specific to MOF materials, and could be discussed in the context of the broader literature relating adsorption to metastability of fluid and to deformation.

For example:

- 1. Grosman, A.; Ortega, C. Influence of elastic strains on the adsorption process in porous materials: an experimental approach. Langmuir 2009, 25, 8083–8093.*
- 2. Grosman, A.; Ortega, C. Cavitation in Metastable Fluids Confined to Linear Mesopores. Langmuir 2011, 27, 2364–2374.*
- 3. Bossert, M., Grosman, A., Trimaille, I., Noûs, C. and Rolley, E., 2020. Stress or strain does not impact sorption in stiff mesoporous materials. Langmuir, 36(37), pp.11054-11060.*

We have added a discussion section to address these points between the lines 210-222, 270-274 and 277-278. An important aspect is to distinguish *elastic* and *inelastic* deformation (phase transition) of the solid. The aspects of elastic deformation are analogous for MOFs and other mesoporous materials. We realize that this aspect also leads to some confusion in the terminology of “rigidity”. One has to clearly distinguish aspects of stiffness (elasticity) in the elastic regime vs. the inelastic regime. In our contribution we mainly discuss differences in flexibility in terms of inelastic deformation (DUT-49: “flexible”, DUT-149 “rigid”) while the unit cell changes in the elastic regime (before the phase transition) are within the limits of error identical.

In this context, and according to the expanded section describing the diffusion hysteresis as proposed by the Reviewer 1, the previously written sentence *"deeper understanding of fundamental differences in contributing diffusion mechanisms leading to observed deviations requires further investigation."* has been removed.

The manuscript mentions MCM41 materials as "rigid", while Fig 2 shows the same order of magnitude of strain during adsorption of alkanes as was previously shown for alkanes in MCM41/SBA15 e.g. in:

4. Gor, G.Y., Paris, O., Prass, J., Russo, P.A., Ribeiro Carrott, M.M.L. and Neimark, A.V., 2013. Adsorption of n-pentane on mesoporous silica and adsorbent deformation. Langmuir, 29(27), pp.8601-8608.

and not only the magnitude but also the trend similar to other non-MOF materials:

5. Gor, G.Y., Huber, P. and Bernstein, N., 2017. Adsorption-induced deformation of nanoporous materials—a review. Applied Physics Reviews, 4(1), p.011303.

We appreciate this suggestion, which clearly confirms our findings. Following sentence was inserted on page 6, lines 142-144 (see also comment above): “These observations are consistent with earlier studies of Gor and co-workers reported on adsorption-induced deformation in SBA-15 and MCM-41 upon physisorption of *n*-pentane³⁸ and described the fundamentals of this technique.³⁹”

The manuscript also lack the details on simulations. It says that Supporting Info has in particular "Details of the ... MD simulations", but it does not. I recommend that not only the details of modeling methods are included in the PDF, but also all the main input files are included, so that the study could be easily reproduced if needed.

We apologize for failing to add the details of the GCMC and MD simulations. The relevant details have been added to the supporting information and representative input files can be found at the GitHub repository: <https://github.com/jackevansadl/supp-data/tree/master/XX-Walenszus>. The code-free link is provided in the “DATA AVAILABILITY” section (lines 283-286). Additionally, we have provided the supplementary crystallographic data for DUT-149 empty framework and crystal structures with defined *n*-butane loadings in this section (lines 286-288).

Additionally, I have a few minor comments:

1. The abstract has a typo "soprtion"

Corrected

2. The role of Fig 1e (top) is unclear, furthermore this figure is not mentioned in the text.

The Fig. 1e illustrates structural configurations of DUT-49(Cu) and DUT-149(Cu) below and above the characteristic pressure p_{NGA} at which of DUT-49(Cu) undergoes transition from the *op*-state into the *cp*-state (and DUT-149(Cu) not). The figure is mentioned in the line 92, modified for clarity, and the corresponding clarification is added to the figure caption.

3. Size of the panels in Fig 2 are not the same

This is due to additional scales on the right side representing the unit cell parameters sizes in Figs. 2c and 2d. To keep the actual size of all figures consistent, the panels were reduced.

4. Different units are used for pressure in Fig 2 and Fig 4

Corrected

5. Figures S7-S15 look like screenshots of very poor quality, I would suggest using the data and preparing the plots which meet the standards of the journal

All previous Figures S7-S15 were replaced by the ones prepared in a better quality

REVIEWERS' COMMENTS

Reviewer #1 (Remarks to the Author):

The authors addressed my comments and questions.

Reviewer #2 (Remarks to the Author):

With the additional discussions, the authors clarified the new understanding gained by the present study comparing to their previous work. My concerns over the novelty of this work were fully addressed, and I recommend the publication of this manuscript in its current form.

Reviewer #3 (Remarks to the Author):

The authors took into account all my comments. I think that these changes, and the changes made based on the comments of other reviewers, made it a better paper. The results were great from the beginning.